# Anaemia in Lambs Caused by *Mycoplasma ovis*: Global and Australian Perspectives

**DOI:** 10.3390/ani12111372

**Published:** 2022-05-27

**Authors:** Peter A. Windsor

**Affiliations:** Sydney School of Veterinary Science, The University of Sydney, Camden, NSW 2570, Australia; peter.windsor@sydney.edu.au

**Keywords:** *Mycoplama ovis*, haemoparasites, sheep, diagnosis, risk factors, management

## Abstract

**Simple Summary:**

*Mycoplasma ovis* (formerly *Eperythrozoon ovis*) inhabits red blood cells and may cause their destruction, leading to anaemia, jaundice and death mainly in lambs, and condemnation of jaundiced carcases at abattoirs. Mycoplasmosis is spread during high-risk procedures that expose or share blood, especially when blood sucking flies and other insects are present on wounds that transfer infection. High-risk procedures include vaccination (re-use of needles), ear-tagging (ineffective disinfection), surgical castration and mulesing (in Australia), and potentially crutching and shearing, with outbreaks usually occurring up to 6 weeks later. Affected animals are weak, lagging in the ‘tail of the mob’ and collapsing when mustered. The investigation of other causes of anaemia and jaundice is required, particularly haemonchosis (Barbers pole worms) and malnutrition. The diagnosis involves the demonstration of *M. ovis* in blood smears and/or by PCR, although the absence of the parasite in smears from affected animals requires the examination of in-contact healthy animals. Treatment with antibiotics is ineffective. For its control, it is required that risky procedures are avoided during high insect activity and the yarding of stock within the next 6 weeks is minimised. Recent anecdotal observations suggest that improved farm practices, including fly control and pain/antiseptic wound dressing may potentially decrease *M. ovis* risk on some farms in some areas of Australia.

**Abstract:**

*Mycoplasma ovis* (formerly *Eperythrozoon ovis*) is a haemotropic parasitic bacterium found within erythrocytes and distributed widely in global sheep and goat production regions. *M. ovis* is transmitted by biting flies and by contaminated instruments, causing morbidity and mortalities from anaemia, usually within 6 weeks following blood-exposure procedures, particularly vaccination, castration, ear tagging, mulesing, and occasionally crutching and shearing. Affected animals develop haemolytic anaemia and may have jaundice, causing abattoir condemnations. The typical history, clinical and pathological findings, display of *M. ovis* in blood smears and/or by PCR is diagnostic, although immune responses deplete *M. ovis* from smears; hence, in-contact healthy animals should be examined. Differential diagnoses include haemonchosis, fasciolosis, malnutrition (copper or vitamin B12 deficiency), and plant toxicities. *M. ovis* parasitaemia may persist, with recrudescence following stressful events, although most older sheep remain immune. Human infections have been reported. Inadequate socioeconomic data present difficulties in assessing the impact of *M. ovis* on production and as antimicrobial therapy is ineffective, its control requires management practices that minimize the impact of invasive procedures in periods when risks of *M. ovis* transmission are more likely. Although considered an emerging pathogen, recent improvements in welfare attitudes and husbandry practices on Australian sheep farms may potentially limit the transmission of *M. ovis.*

## 1. Introduction

*Mycoplasma ovis* belongs to a highly specialized group of bacteria with a unique cell tropism to erythrocytes that was reclassified to the genus Mycoplasma as the so-called haemotrophic mycoplasmas (also named hemoplasmas) at the beginning of the twentieth century [1]. The haemotrophic mycoplasmas of livestock comprise *M. suis*, *M. parvum*, and ‘*Candidatus M. haemosuis*’ in pigs, *M. wenyonii* and ‘*Ca. M. haemobos*’ in cattle, as well as *M. ovis* and ‘*Ca. M. haemovis*’ in sheep and goats [1,2]. The clinical outcome of haemoplasma infections is highly variable, ranging from life-threatening anaemia to various chronic syndromes (e.g., mild anaemia, poor performance, reproductive disorders) or even asymptomatic courses. The differential diagnosis of anaemia in small ruminants should consider the inclusion of the haemoparasite *M. ovis*, formerly known as *Eperythrozoon ovis* and considered as a Rickettsia in the family Anaplasmataceae, until molecular studies reclassified the organism as a Mycoplasma [1].

Haemotropic *M. ovis* occurs almost worldwide in small ruminant populations, also affecting a broader range of mammalian hosts, including deer, reindeer and occasionally, humans [2]. Field observations indicate that the mechanical transmission of *M. ovis* occurs both from the bites of haematophagous arthropods provided with access to open wounds, and iatrogenic infection from contaminated sharp instruments. Acute mycoplasmosis typically causes severe haemolytic anaemia and mortality in young animals. Chronic *M. ovis* infections may produce mild anaemia and varying degrees of morbidity, depending on a range of host–pathogen–environment (HPE) factors that include age, level of nutrition, immunological and reproductive status and the presence of concurrent infections.

Haemotropic *M. ovis* is often considered endemic in major sheep production areas, although there are few published reports of prevalence and socioeconomic studies of the impacts of the disease [3]. Of interest is that there are increasing reports of human infections with haemoprotozoa, particularly in pregnant women, immunocompromised patients and people regularly exposed to animals and arthropods. Furthermore, there is the challenging diagnostic issue of the reliance on microscopic evaluation of Giemsa-stained blood smears, although *M. ovis* is often absent in severely anaemic animals. PCR diagnostics are increasingly required, yet reliable published reports of phylogenetic trees and whole genomic studies are lacking. Finally, the likely transboundary movement of *M. ovis* within infected small ruminants, along with other parasitic, bacterial and viral infections to uninfected areas, risks compromising the increasing global trade of small ruminants [4]. A review of the available information on *M. ovis* as a disease of small ruminants and potentially other species that is increasingly considered an emerging pathogen and deserving of increased attention, is appropriate.

## 2. Epidemiology: Transmission, Distribution, Prevalence and Zoonotic Potential

### 2.1. Transmission of M. ovis

*M. ovis* is an obligate epicellular bacteria that is mechanically transmitted by haematophagous arthropods that occur in the vicinity of livestock, including flies (e.g., *Stomoxys calcitrans*, *Haematobia irritans*, *Tabanus bovinus*, *T. bromius*, *Melophagus ovinus*), midges (e.g., *Culicoides brevitaris*) and mosquitoes [2,5]. Importantly for Australia, this also suggests the likelihood of transmission by *Lucilia cuprina*, the major cause of ovine myiasis [6]. Molecular studies have provided evidence of mechanical transmission by a range of species of ticks (e.g., *Amblyomma*, *Hyalomma*, *Rhipicephalus* and *Haemaphysalis*) [2] and lice [7]. As arthropod vectors are considered essential in the epidemiology of haemoplasmas, seasonal changes in arthropod density and distribution influence the prevalence of *M. ovis* infection of sheep in Australia [8]. Similarly, the presence of ticks on small ruminants is associated with haemoplasma-positive status and disease severity [9]. High levels of biting activity are considered essential for vector-borne transmission where low levels of parasitaemia occur, with the minimum infective dose of *M. ovis* suggested as a single parasitised erythrocyte [10]. Although mechanical arthropod-borne transmission appears to occur, it is uncertain if a cyclical transovarial process is involved. Further, the heavy blood-feeding by some arthropods may contribute to losses of blood, increasing the severity of anaemia. Tropical and subtropical temperatures, rainfall and humidity are favourable for the propagation of haematophagous arthropods and account for both the high prevalence of vector-borne diseases in these regions and the more susceptible populations in temperate and cool climate zones, where the occurrence of fly and tick vectors is more sporadic, so immunity to *M. ovis* more variable [2]. The possibility of transplacental transmission of haemoplasma infection in cattle has been suggested, although it is uncertain whether *M. ovis* infects reproductive tissues and transplacental transmission during pregnancy occurs in small ruminants [11]. Iatrogenic transmission through contaminated needles, especially by vaccination ‘guns’, ear tag applicators and equipment used in crutching, shearing and mulesing, is also considered important in the epidemiology of *M. ovis* in small ruminant flocks [12].

### 2.2. Distribution of M. ovis

*M. ovis* occurs in sheep and goats in the tropics, subtropics and in many temperate zones and is considered an emerging pathogen [2,12]. The disease has been reported in Africa, Australia, Europe, North and South America, and in Southeast and East Asia, with reports from China, Japan, Malaysia and the Philippines. However, there appears to be considerable variation in prevalence in the different parts of the world that very likely reflects a range of spatial (e.g., environmental variability) and temporal (e.g., age and exposure susceptibility) factors, although it may also reflect the different diagnostic and survey methods used [2]. In an Australian serosurvey using IFAT, a seroprevalence of 49% was recorded in chronically infected adult sheep in the state of Tasmania [13]. A systematic survey of 20 weaner sheep from each of the 91 farms in the state of Western Australia using ELISA detected an overall seroprevalence of 4.5% of sheep for 47% of the farms sampled, although the serological evidence of infection was highly variable between agricultural regions, varying from 79.5% in the south-east to 12.8% in the northern region [14]. Molecular studies of *M. ovis* using PCR and sequencing also indicate the highly variable prevalence between region and survey approaches, from 6.3% as determined in Tunisia [15] to 100% infection rates occurring in small ruminants in Mexico [16]. The numerous countries where outbreaks of *M. ovis* infection have been reported, has been reviewed [2].

### 2.3. Zoonotic Potential of M. ovis

Haemotropic mycoplasma infections in humans are likely to have been under-diagnosed. An overall seroprevalence of 35.3% of human haemotropic Mycoplasma infection, including 57% and 100% infection rates in women and their new-born babies with mostly mild signs, was reported from inner Mongolia of PRC [17]. Molecular studies have reported *M. ovis*-like, *M. haemofelis*-like, *M. haemominutum*, *M*. *haematoparvum* and *Ca. M. haemohominis* infection in humans [2,18], with persistent co-infection of *M. ovis* and *Bartonella henselae* reported in a veterinarian with a history of protracted illness and non-specific signs [5]. These findings suggest that haemotropic mycoplasmosis may be an emerging zoonotic concern, particularly as an occupational health and safety concern for veterinarians, veterinary support staff and students, herdsmen, farmers, pastoral community members and potentially wildlife workers, where exposure to animals and the arthropod vectors of haemoplasma may commonly occur [17,18]. Further, the risk of human haemotropic mycoplasmosis is also increasingly recognised amongst HIV patients [5,18].

## 3. Pathogenesis: Clinical Signs, Pathology and Diagnosis of *M. ovis* Infection

### 3.1. Pathogenesis of M. ovis Infection

Following entry by mechanical or iatrogenic transmission, *M. ovis* multiplies primarily in the bone marrow. The incubation period is between 5 and 12 days post-infection (pi) prior to appearing as a parasitaemia, with peak levels of parasitaemia and anaemia reported between 8 and 30 days pi in experimentally infected sheep [19,20]. The incubation period of *M. ovis* in experimentally infected sheep appears to be inversely proportional to the size of the infectious dose, with shorter incubation periods in sheep experimentally infected with heavily parasitised blood and more extended incubation periods after inoculating blood with low levels of *M. ovis* [2,10]. The level of parasitaemia developing during *M. ovis* infection in small ruminants reflects the percentage of parasitised erythrocytes and has been described as: mild (1–29%); moderate (30–59%); or severe (>60%) [21]. Altered haemodynamics, due to the deformation of erythrocyte membranes and increased membrane fragility, lead to erythrophagocytosis and haemolytic anaemia [21,22]. A significant decrease in circulating erythrocyte numbers depresses the haematocrit and haemoglobin concentrations, with haemolysis associated with haemoglobinuria, bilirbinaemia and deposition of haemosiderin in tissues [23].

HPE interactions that determine the clinico-pathological course of diseases are critical in understanding the expression of *M. ovis* infections. The relatively low pathogenicity of *M. ovis* in sheep is indicated by the observations that uncomplicated infections are typically asymptomatic. However, where the host animal is young or immunocompromised, and the environment is conducive to pathogen transmission, as occurs with the presence of arthropod vectors or use of husbandry practices that induce iatrogenic infections, outbreaks of acute severe haemolytic anaemia may well occur in spite of the relatively low pathogenicity of *M. ovis* in sheep. Host immune reactions are integral to the development of acute and chronic *M. ovis* infections, with humoral immunity producing anti-erythrocyte antibody, injury and agglutination of erythrocytes, then phagocytosis by Kupffer cells of the liver and reticular cells in lymphoid tissues [2]. Further, oxidative injury, disruption of cell functions, immune evasion and secretion of lytic enzymes by haemoplasmas that also contribute to the development of haemolytic anaemia, have been described [24].

The common observation from field studies that previously infected small ruminants were resistant to subsequent challenge by *M. ovis* was confirmed in experiments and indicates that robust immune responses to infection occur [19]. The presence of circulating antibodies within 1 to 2 weeks post-infection confers resistance and inhibits reinfection, suppressing parasitaemia and prolonging the prepatent period of infection in passively immunised sheep [2]. It was also observed that splenectomised animals became susceptible to infection. This confirmed the active role of the spleen in the development and maintenance of resistance in sheep, with splenic macrophages and reticular cells responsible for the clearance of parasitaemia by the physical detachment of *M. ovis* from erythrocyte membranes [2]. The evasion of the immune system and the establishment of persistent infections are key features in the pathogenesis of livestock mycoplasmas, suggesting knowledge of the underlying mechanisms of this phenomenon may provide insights for the development of therapy and prophylaxis strategies against Haemotropic mycoplasma infections [25]. The induction of persistent infection is considered central to high-level adaption, host-specificity and survival of *Mycoplasma* spp. Understanding the molecular pathways of the HPE interactions, including pathogen persistence, is suggested as important for mycoplasmal research. Next-generation high-throughput technologies enabling genomics, proteomics, transcriptomics and metabolomics tools may well assist in elucidating the pathobiology and host responses involved in mycoplasma-associated livestock diseases [25].

### 3.2. Clinical Signs of M. ovis Infection

Although *M. ovis* infections may occur in all age groups of small ruminants, profound haemolytic anaemia, icterus, haemoglobinuria, reduced exercise tolerance and lowered weight gain and mortality are most frequently encountered up to 6 weeks post-marking in lambs and weaner sheep [12,26], although there are anecdotal observations that the occurrence of these outbreaks may be increasingly sporadic in Australia. The severity of anaemia in acute disease varies among susceptible individuals in a flock, with age, nutritional and immunological status, and the presence of concurrent infections of relevance to disease expression. Poor body condition scores from stunted growth, delayed attainment of sexual maturity, decreased exercise tolerance, abdominal distension, emaciation and pallor are features of chronic disease in lambs.

Due to the development of acquired immunity to *M. ovis*, clinical disease is less commonly observed in adult animals. It is likely that some animals with chronic *M. ovis* parasitaemia do suffer mild regenerative anaemia under field conditions, although most are likely to remain undetected in sheep flocks in extensive pastoral conditions. Some of these adult sheep with chronic *M. ovis* may occasionally succumb to acute infection and develop haemolytic anaemia, most often associated with the stress of pregnancy, parturition, handling, malnutrition and concurrent parasitic, viral or bacterial infections and other immunosuppressive conditions. Mild anaemia, ill thrift, reduced weight gain, jaundice, exercise intolerance, decreased wool and milk production are described features of chronic disease in adult sheep and goats.

### 3.3. Pathology of M. ovis Infection

Lesions observed in fulminating *M. ovis* infections in sheep vary considerably depending on the degree of infection severity and clinical duration. At necropsy, typical lesions include pallor and/or jaundice of the mucous membranes, thin watery blood, marked splenomegaly from congestion (Figure 1) and with a prominence of the white pulp, haemoglobinuria and enlarged kidneys that have a distinctive brown discolouration due to deposition of haemosiderin. In addition, the gallbladder is distended with bile and there is often evidence of hypoproteinanemia observed as clear to icteric transudates in body cavities. The common histological lesions associated with *M. ovis* infection include the enlargement of the renal Malpighian corpuscles, haemosiderosis of the spleen and kidneys, moderate periacinar necrosis in the liver, depletion of lymphoid tissues in the spleen and lymph nodes and hyperplasia of haemal lymph nodes [23].

### 3.4. Diagnosis of M. ovis Infection

As *M. ovis* is an obligate epicellular bacterium that is unstable in vitro and is readily destroyed by drying or exposure to disinfectants, it is unable to be grown on cell-free media. Although *M. ovis* was successfully passaged through the yolk sac of embryonated chicken eggs and maintained by its attachment to erythrocytes in heparinised samples incubated in supplemented Eagle’s medium under 5% CO_2_, a suitable method for the in vitro cultivation of haemotropic Mycoplasmas in the laboratory is yet to be established [2]. Thus, the diagnosis of *M. ovis* presently relies on detailed history, clinico-pathological evidence, dismissal of differential diagnoses, and direct examination laboratory analyses. The microscopic evaluation of stained blood smears, haematological and biochemical analyses and histopathology are the most commonly used tools for routine *M. ovis* diagnosis, although serology for antibodies and PCR assays are increasingly in use, particularly for prevalence and genomic studies, respectively. Despite the current advances in genotyping and molecular proteomics of various parasitic pathogens and the apparent global emergence of haemotropic mycoplasmosis as an economic concern to small ruminant producers, comprehensive reports are lacking on the genomics of haemotropic *M. ovis* [2].

Microscopic examination of blood smears stained with Romanowsky dyes is a rapid and inexpensive diagnostic method of diagnosis. Observed under the light microscope, smears containing haemoplasmas are recognised by the binding of organisms to the surface of erythrocytes (Figure 2), although they may be found lying loosely in the plasma due to detachment from cells, especially after prolonged storage of blood samples [27]. *M. ovis* is identified as basophilic pleomorphic (coccoid, coco-bacillary, ring, dumb-bell or horseshoe-shaped) bodies measuring approximately 0.3–1 μm in diameter, either singly or in short chains on the erythrocytes or as free bodies in the plasma [20]. Challenges to the microscopic examination of smears include the differentiation of *M. ovis* from stain deposits, cell fragments and other artefacts, and the limited sensitivity and specificity in the microscopic diagnosis of haemotropic mycoplasmosis because of cyclic parasitemia and the prevalence of mild infections with low parasitaemia [27]. The use of more advanced microscopic techniques, including fluorescent, confocal and scanning electron microscopes, provides more definition of *M. ovis* morphology [2], although microscopy remains far less specific than the molecular detection methods that provide in excess of 90% diagnostic specificity [28].

The serological detection of antibodies by a specific IFAT (indirect fluorescent antibody test) for the detection of ovine and bovine haemotropic Mycoplasmas enabled a robust serodiagnostic approach for prevalence surveys of *M. ovis* in sheep in Australia [14]. Other serological tests used in trials include the Coomb’s test, a CFT (complement fixation test) using antigens prepared from lysed red blood cells, and an ELISA (enzyme-linked immunosorbent assay) that remains in use as a confirmatory test for the diagnosis of *M. ovis* infection in small ruminants [2]. The ELISA is now the preferred serological test for prevalence surveys of small ruminant flocks and herds, although is of limited value in routine diagnosis as antibodies to *M. ovis* are transient.

The molecular detection of *M. ovis* by PCR and sequencing of the 16S rRNA gene is increasingly used to detect and characterise haemotropic Mycoplasmas in animal and human infections, improving laboratory diagnostics and assisting in elucidating species diversity and host ranges [2,5]. It was a PCR analysis of the 16S rRNA sequence of *E. ovis* that confirmed the phylogenetic relationships with the genus *Mycoplasma* and helped to unravel novel species and host adaptations, including reports of the detection of the *M. ovis* genome in captive cervids and free-ranging deer in Brazil [2,5]. Further, PCR enabled the detection of *M. haemofelis*, *M. suis* and *M. ovis* [29] and *Ca. M. haemohominis* in humans [2,5]. Real-time PCR assays are now available for evaluating the significance of a positive PCR result and monitoring the course of the disease. Real-time PCR has been used for the direct quantification of haemotropic Mycoplasma DNA, used in studies of transplacental and vector-borne transmission of bovine infections and in surveys to determine the prevalence and risk factors of haemotropic Mycoplasmas in companion animals [2]. Comparative genomic analyses are improving knowledge of the genetic basis of virulence by predicting the potential virulence factors of parasites, although currently, there is no published information on the molecular basis of virulence in *M. ovis* infection [2].

### 3.5. Concurrent Infections and the Differential Diagnosis of M. ovis

The co-infection of erythrocytes with multiple species of vector-borne *Mycoplasma* spp. and a range of haemoprotozoa is a not uncommon finding in livestock in some countries. Reports of haemotropic *Mycoplasma* as a co-infection with haemoprotozoa include the following: with *Anaplasma* sp. in sheep and goats from Morocco; with *Babesia* and *Theileria* spp. in sheep flocks from Turkey; with *Ehrlichia ewingii* in a goat from the USA with macrocytic hypochromic anaemia; and with *Anaplasma*/*Babesia* spp. in association with the increased severity of anaemia in chronic disease of small ruminants in the USA [2]. It has also been suggested that co-infection with tick-borne haemoprotozoa, including *Theileria*, *Babesia*, *Anaplasma* and *Ehrlichia* spp., increases the susceptibility of animals to haemotropic mycoplasmosis [29,30]. The presence of multiple co-infections of haemopathogens likely provokes a complex pathological process that enables synergy of infection and more successful colonisation of erythrocytes, potentially leading to haemolytic anaemia [2].

Concurrent endoparasitism, due to *Haemonchus contortus* and possibly other pathogenic *Strongyle* spp., is very likely to increase the severity of anaemia and pathology of haemotropic mycoplasmosis in small ruminants [31]. *H. contortus* is considered the most important internal parasite of sheep and goats and is capable of extreme pathogenicity. Adult worms suck blood from the mucosa of the abomasum, causing severe anaemia, weight loss, decreased milk production, compromised body condition scores and interruption of wool growth (so-called ‘wool break’), often accompanied by submandibular and cervical soft-tissue and abdominal cavity transudates due to hypoproteinaemia (so-called ‘bottle jaw’). Mortalities frequently occur and a definitive diagnosis requires the examination of the abomasum to display the characteristic parasites, preferably accompanied by a total worm count of the contents of the abomasum.

Whilst parasitic gastroenteritis (PGE) and haemotropic mycoplasmosis are both associated with anaemia and weight loss in small ruminants, the mechanisms involved are very different. In PGE, blood is lost into the gastro-intestinal tract of the parasite, whereas in clinical *M. ovis* infections, blood loss occurs in the vascular system and spleen. Thus the patho-morphology of PGE and haemolytic anaemia are usually readily distinguishable, although the possibility that they are present together in the same animal should be considered. Of interest is the consensus that haemotropic *Mycoplasma* spp. may act synergistically with highly pathogenic nematodes and particularly *H. contortus*, contributing to the severity of disease in concurrently infected flocks [32]. The relevance and contribution to the severity of anaemia from haemotropic mycoplasmosis in concurrent infections is almost certainly influenced by the metabolic status of the host. Malnutrition and the transient immunosuppressive events aligned with pregnancy, lactation and parturition are all very likely to increase the severity of disease in small ruminants [23]. Observations that splenectomy or the daily administration of dexamethasone induces the signs of haemolytic anaemia in experimental haemotropic mycoplasmosis in animal models are consistent with the suggestions that concurrent infections enhance the severity of *M. ovis* in small ruminants through immunosuppressive events [2,5].

In addition to PGE, other pathological entities that cause clinical anaemia are relevant to the differential diagnosis of *M. ovis* infection. In the Australian sheep production context, this includes a broad range of diagnoses, including copper deficiency, especially in animals grazing pastures on porous volcanic soils, or cobalt deficiency leading to inadequate ruminal synthesis of vitamin B12 deficiency, especially in animals grazing pastures on limestone-derived soils. Also of relevance to the differential diagnosis in young animals is hypoproteinaemia from malnutrition and plant intoxications causing haemolysis (e.g., in animals grazing on copper accumulating pastures), particularly in drought and post-drought periods, respectively, with other causes of haemoparasitism also possible [20,33]. In older animals, fasciolosis, malnutrition, plant poisonings affecting liver function and causing jaundice including exposure to copper accumulating and pyrrolizidine alkaloid containing plants on pasture, copper toxicity and other causes of haemoparasitism, may be considered [20,33]. Recently in Spain, clinical anaplasmosis in sheep with jaundice and carcass condemnation at abattoirs involving *Anaplasma ovis* infection has been identified as causing severe disease, despite anaplasmosis being an historically neglected entity previously considered to induce only mild clinical signs [34].

## 4. Discussion

This review of the currently published literature of haemotropic *M. ovis* indicates that it is an endemic infection in many small ruminant populations globally and considered an emerging and potentially zoonotic pathogen, following detections in deer, reindeer and humans. The microscopic examination of stained blood smears was the earliest method of diagnostic detection, although complicated by the need to avoid or be very careful in mustering clinically affected animals with anaemia, with a preference for sampling unaffected individuals where the likelihood of demonstrating parasitised erythrocytes is optimised. Serosurveys have been commonly conducted with vastly different prevalence rates identified, reflecting the HPE interactions and spatial and temporal factors involved in *M. ovis* disease expression. More recently, the characterisation of *M. ovis* parasitaemia in sheep and goats has been conducted using various PCR assays. Unfortunately, published socioeconomic studies of the impacts of the disease are lacking and this confounds obtaining research funding from livestock industry research development agencies to address the many research gaps in our knowledge of *M. ovis.*

Priority areas for future work on *M. ovis* are suggested to include genomic analyses, addressing challenging diagnostic issues including studies of cell-mediated immune response mechanisms, and increasing investigations of humans with potential zoonotic infections. Further, the conduct of risk analyses to examine the likelihood of increasing transboundary transfer of *M. ovis* accompanying other parasitic, bacterial and viral pathogens to uninfected areas is suggested, addressing concerns on pathogen compromise of the global trading of small ruminants [4]. An emerging issue of increasing concern is that climate change impacts are increasing the arthropod vector ranges in many parts of the world and this could contribute to the spread of *M. ovis* and haemoparasite infections to currently uninfected host populations [34]. Finally, improving our understanding of the mechanisms of persistent infection is of interest, potentially offering pathways for developing novel vaccination strategies and therapeutics, has been suggested [25]. However, there is an urgent need to reduce the use of antibiotics in farm animals. With the general lack of efficacy of the therapeutics investigated for use in *M. ovis* outbreaks [23,26,35,36] and promotion of the importance of minimizing the handling of affected animals other than gently moving the mob to a paddock with good feed and water and leaving them alone for 4 to 6 weeks to enable their recovery from disease [26], the use of therapies is not recommended. Disease control is focused on managing the risk factors that enable the transmission of *M. ovis.* An understanding of management interventions required to limit pathogen expression where therapeutic and prophylactic options are unavailable, is critical.

This review of mostly peer-reviewed published reports confirms that *M. ovis* is historically a pathogen of concern in the Australian sheep industry. Outbreaks were regularly investigated prior to the new millennium, with numerous experimental and survey studies reported [8,10,12,13,14,20,22,34], including mortalities in lambs as high as 6% [35]. Only two reports, neither peer-reviewed, were identified from the new millennium [26,36]. Of considerable interest is that recent anecdotal observations within some parts of Australia suggest the occurrence of *M. ovis* outbreaks may have declined in recent years. Few *M. ovis* outbreaks are currently reported to animal health authorities or concerns discussed with sheep veterinarians, with the likelihood that this decline is a valid observation that may be linked to changing sheep demographics and on-farm attitudes and practices. There has been a decrease in the population of Merino sheep as a proportion of the national sheep flock in the last few decades, particularly with an extended period of low wool prices and rapidly rising demand for exported sheep meat products. The percentage of Merino breeding ewes fell 3% in the year 2021 and now accounts for under 75% of all breeding ewes [37], despite projections that sheep numbers will grow by 4.9% to 74.4 million head in 2022, reaching its highest level since 2013 [38]. Far fewer sheep that still require the mulesing operation to provide lifetime protection against myiasis are now present in Australia [6,39,40,41,42]. This follows the increasing uptake of Merino rams selected for less breech skin wrinkle and susceptibility to flystrike and the widespread adoption of cross-breeding and importation of meat breeds for the expanding export trade in sheep meat trade.

Of likely relevance to the observations of a potential decline in *M. ovis* outbreaks has been the significant improvement of hygiene and animal welfare practices on many sheep production enterprises in Australia [42]. This includes the mandatory accreditation for those performing the mulesing operation and increasing use of professional contractors for implementing the husbandry procedures required at lamb marking and shearing. This change management has likely contributed to a decline in the sharing of blood-soaked surgical husbandry intervention instruments, increased use of antisepsis and more appropriate administration of blowfly control products, providing prolonged control of arthropods (dicyclanil; CLiK™ EXTRA, Elanco, Macquarie Park, NSW, Australia). Further, since pain welfare research that commenced in 2005, the Australian sheep industry has experienced the widespread availability of a farmer-applied wound pain mitigation spray-on product for lamb marking and mulesing (Tri-Solfen^®^, Medical Ethics Ltd., Melbourne, Victoria, Australia). This product contains two local anaesthetics, adrenalin and the antiseptic cetramide. With a pH of ~2.8, it appears the product also provides viricidal and antibacterial activity [43,44] that may contribute to reduced *M. ovis* transmission at marking, mulesing and shearing.

Improving sheep producer welfare attitudes and practices, including pain management during aversive husbandry procedures that is now commencing globally, has been documented for well over a decade in Australia [39,40,41,42,43]. Selecting animals that are adapted to local conditions is recognised as reducing livestock welfare problems, particularly those in extensive systems, with practice-led innovations enabling the ownership of issues by producers and the implementation of improved knowledge-exchange strategies [45]. These improvements to animal health practices in the Australian sheep industry were observed to have commenced as a response to the ovine paratuberculosis crisis that occurred in the late 1990′s and into the new millennium, when substantial mortalities from ovine Johne’s Disease (OJD) occurred [46]. The OJD crisis resulted in the rapid introduction of widespread vaccination (Gudair^®^, Zoetis, Rhodes, NSW, Australia) that resolved the major socioeconomic crisis that had developed throughout the wool sheep industry [46]. This unsettled period was soon followed by the mulesing crisis, when animal welfare activism targeted Australian Merino wool production as involving cruel practices. This led to the rapid introduction of farmer-applied pain management products and improvement of on-farm welfare attitudes and practices [39,40,41,42]. It has been suggested that a new paradigm has emerged on farms, capable of sustainably addressing the complex welfare concerns arising in extensive livestock husbandry systems in Australia [40]. Increasing consumer awareness of these improving attitudes and practices is required [42]. If the apparent decline in *M. ovis* outbreaks is a valid observation, it is tempting to suggest that these positive attitudinal and practice changes in the Australian sheep industry may have contributed to a reduction in the risks of transmission of *M. ovis,* and thus outbreaks, in spite of climate change impacts that may increase the risk of arthropod transmission of the pathogen.

## 5. Conclusions

Haemotropic *M. ovis* is endemic in many small ruminant populations, although there are few published studies of the socioeconomic impacts of the disease or genomic analyses. With increasing evidence of zoonotic infections, persistence of challenging diagnostic issues and the likelihood of transboundary transfer of *M. ovis*, along with other parasitic, bacterial and viral infections to uninfected areas, there are risks of *M. ovis* compromising the increasing global trade of small ruminants. An understanding of the mechanisms of persistent infection has been suggested as potentially providing novel vaccine strategies and therapeutic options [25]. However, with the increasing importance of reducing the usage of antibiotics in farm animals and as no effective therapeutics have been identified for *M. ovis*, the need for disease control by flock and herd management is indicated. Recent anecdotal observations from Australia suggest that the occurrence of *M. ovis* outbreaks may have declined in recent years. The hypothesis is proposed that if this observation is confirmed, it may be linked to improving on-farm animal health and welfare practices, particularly with the increasing use of prolonged fly control and pain mitigation products for lamb marking, mulesing and shearing that likely reduce *M. ovis* transmission risks. Applied research seeking evidence that may inform this hypothesis and work to address the absence of knowledge on the socioeconomic impacts of this enigmatic disease is suggested as appropriate.

## Figures and Tables

**Figure 1 animals-12-01372-f001:**
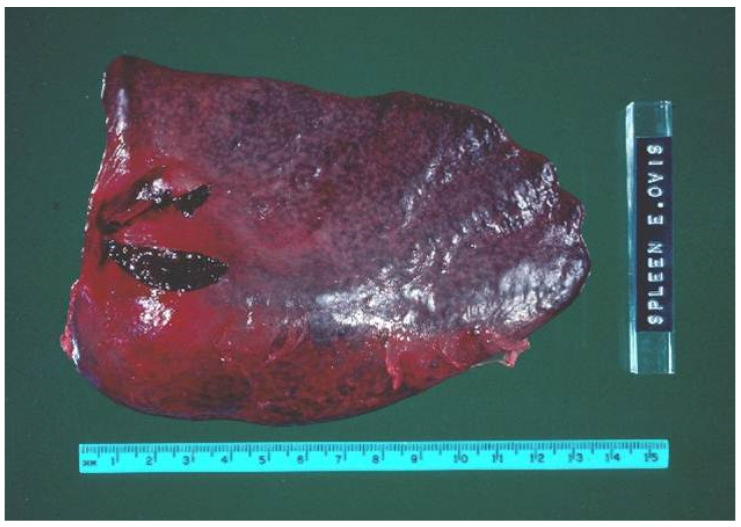
Historical image of splenomegaly in a lamb diagnosed with *Epyrethrozoan ovis*, now *Mycoplasma ovis*.

**Figure 2 animals-12-01372-f002:**
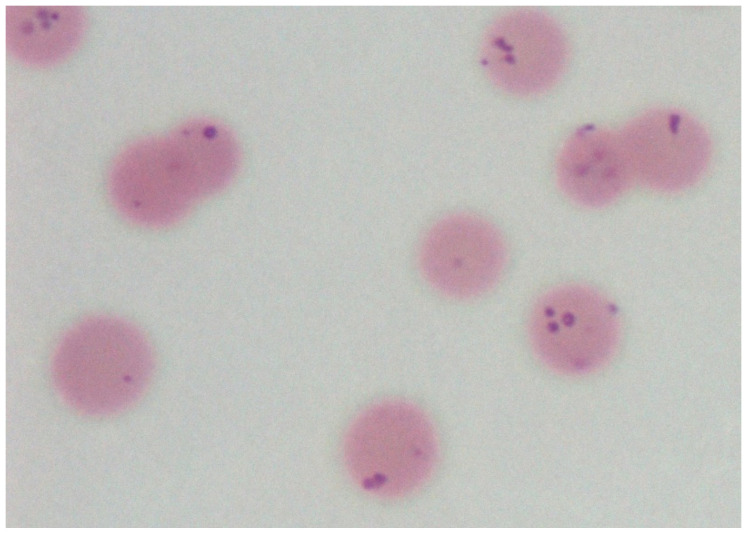
Image from an ovine blood smear smear containing erythrocytes parasitised by basophilic bodies, consistent with the morphology of *M. ovis*.

## Data Availability

Not applicable.

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
