# Peer review of "Anaemia in Lambs Caused by Mycoplasma ovis: Global and Australian Perspectives"

_animals, 2022, doi:10.3390/ani12111372_

Round 1
Reviewer 1 Report
The manuscript "Anemia in lambs caused by Mycoplasma ovis: global and Australian perspectives" focuses on mycoplasma ovis infection, a sneaky pathogen with zoonotic potential that causes noteworthy health and economic problems in sheep and goat farming in the world. Unfortunately, this pathogen is less investigated, even if the "first reported" cases start being more frequently. This review is well written and clear and highlights an underestimated health and management problem in small ruminants giving food for thought to researchers in the sector, even outside Australia.
For these reason I support the pubblication on "Animals" journal.
However, I would like to bring a few small points to the author's attention:
Line 15: I suggest to change "Barbers Pole" with Haemonchus contortus
Line 112: Cite this reports, please
Line 71: "trrees", correct please
In conclusion, for greater completeness of the review, I suggest adding a short chapter on the molecular characterization of M. ovis.
Author Response
Thank you for this positive review and the suggestions, with amendments as follows:
However, I would like to bring a few small points to the author's attention:
Line 15: I suggest to change "Barbers Pole" with Haemonchus contortus; Yes
Line 112: Cite this reports, please; Yes
Line 71: "trrees", correct please; Yes
In conclusion, for greater completeness of the review, I suggest adding a short chapter on the molecular characterization of M. ovis; Yes.
All incorporated in the resubmitted mansucript.
Reviewer 2 Report
This is a nice review article, the subject is reviewed in deep and the local, regional and international scope is well exposed. The article is clear, well organized and the discussion is pertinent.
Some minor changes:
L9 I suggest avoiding redundance in “(formely Eperythrozoonosis)”
L71 phylogenetic trees
L322 could authors explain in more detail why B12 vitamin deficiency is relevant in the Australian context, for what type of production system is important?
Author Response
Thank you for the positive comments on this paper being clear, well organized and the discussion is pertinent. The minor changes suggested have been addressed in the resubmitted manscript:
L9 I suggest avoiding redundance in “(formely Eperythrozoonosis)”; yes.
L71 phylogenetic trees; yes
L322 could authors explain in more detail why B12 vitamin deficiency is relevant in the Australian context, for what type of production system is important?; yes. This is an important entity in parts of Australia where clinical signs of anaemia and ill thrift due to Cobalt deficiency & Vitamin B12 deficiency occur in sheep (eg pastures where limestone soils exist, especially in South Australia).
Reviewer 3 Report
The review by Windsor P. A. is nicely bridging the knowledge about the former Eperythrozoon ovis and the new M. ovis.
Because literature is rare on this topic, I have a few suggestions to make for clarification or details purpose:
- Antimicrobial treatment success: from L360 and elsewhere in the manuscript, I understood that no treatment is efficient. However for another ruminant hemoplasma species ( wenyonii), tetracyclines have been described as allowing a rapid clinical recovery. Could you please confirm that no treatment is efficient? I am missing a section on this specific topic (treatment and control)
- Zoonotic potential
- L66 “Increasing reports of human infections”: this is a strong statement that needs to rely on several validated references. Please provide references.
- L125 to 132: please specify the clinical signs in the three first mentioned studies. Please specify the immune status of the patients in each study. Were they immunocompromised patients?
- L175-180: What about in humans? Is there any acquired immunity in workers regularly exposed to M. ovis?
- Experimental reproduction of the disease: L148, please be more specific about what is considered a low versus a heavy infectious dose for a non-cultivable bacterium
- Diagnostic: L272-278: please can you be more specific about the interest of quantification: what should be considered as a clinically relevant Cp or Ct ?
- Chronic form and immunity: L207-208, what do we know about immunity in chronic forms?
- Link between animal health and animal welfare in Australia: in two mentioned occurrences, trying to control a disease through vaccination or mulesing, results in transmission of M. ovis. However, good operating practices were enough to control the risk of infection. The Tri-Solfen spray appeared to contribute to both health and welfare. In the end, it is not clear to me how much of the decrease in M. ovis prevalence in Australia is due to improved practices having also an impact on improved welfare, versus lowering the proportion of Merino sheep, etc.
Other minor remarks
L71 trees instead of trees
L87 arthropod (“r” is missing)
Author Response
Thankyou for the comments and where possible, all the suggestions have been addressed as fully as possible. However, as this is an under-researched pathogen, some queries are difficult to provide more specific detail, as follows:
- Antimicrobial treatment success. Response: in our context, use of tetracyclines and other therapies have been attempted and these are cited (eg 26, 36), with an explanation now added of what is recommended; ie leaving them alone as attempts to treat them do precipitate mortalities!
- Zoonotic potential
- L66 “Increasing reports of human infections”. Response: this was reviewed in the paper by Bura et al & the included references in this section, mostly described in reports from Inner Mongolia in the PRC as causing mostly mild disease (ie anaemia).
- please specify the clinical signs: Response: mild anaemia, mostly observed in placental blood. The populations are from undeveloped areas where malnutrition may well be present, although the immune status of the patients in each study is not documented; they do not appear to be immunocompromised patients (as in HIV).
- L175-180: What about in humans? Response: acquired immunity in workers regularly exposed to M. ovis would be expected although there is minimal information available on this.
- Experimental reproduction of the disease: L148. Response: the dose provided is the percentage of infected erythrocytes and describe as mild (1 - 29%); moderate (30 - 59%); or severe (>60%); see Reference 21.
- Diagnostic: L272-278: Response: PCR quantification of M ovis is described by Bura et al and is cited without specific additional information as this is not within local expertise, mostly relying on haematology of in-contacts.
- Chronic form and immunity: L207-208. Response: as we do not see chronic forms of M. ovis in adult sheep Australia, this paper can only cite what others have described, noting the paucity of information on mechanisms of immunity in chronic disease.
- Link between animal health and animal welfare in Australia. Response: As mulesing is a bloody procedure, the risk of transmission of M. ovis is high, both iatrogenic and arthropod-induced. Whether the prevalence in Australia is decreasing and if so, is it due to improved husbandry practices & Tri-Solfen spray (now very widely used), less Merino sheep, or a combination of these and other factors, is speculative at this stage. This question is raised in this paper to suggest future research topics, although it is noted that M ovis is not seen as a priority by sheep industry funding agencies.
Other minor remarks
L71 trees instead of trees: Yes.
L87 arthropod (“r” is missing): Yes
Round 2
Reviewer 3 Report
The revised version in now in a form suitable for publication. Many thanks for your detailed answers to my questions.